# Sporadic Inclusion Body Myositis: An Acquired Mitochondrial Disease with Extras

**DOI:** 10.3390/biom9010015

**Published:** 2019-01-07

**Authors:** Boel De Paepe

**Affiliations:** Neuromuscular Reference Centre, Ghent University Hospital, Corneel Heymanslaan 10, 9000 Ghent, Belgium; boel.depaepe@ugent.be

**Keywords:** mitochondrial dysfunction, myositis, sporadic inclusion body myositis

## Abstract

The sporadic form of inclusion body myositis (IBM) is the most common late-onset myopathy. Its complex pathogenesis includes degenerative, inflammatory and mitochondrial aspects. However, which of those mechanisms are cause and which effect, as well as their interrelations, remain partly obscured to this day. In this review the nature of the mitochondrial dysregulation in IBM muscle is explored and comparison is made with other muscle disorders. Mitochondrial alterations in IBM are evidenced by histological and serum biomarkers. Muscular mitochondrial dynamics is disturbed, with deregulated organelle fusion leading to subsequent morphological alterations and muscle displays abnormal mitophagy. The tissue increases mitochondrial content in an attempt to compensate dysfunction, yet mitochondrial DNA (mtDNA) alterations and mild mtDNA depletion are also present. Oxidative phosphorylation defects have repeatedly been shown, most notably a reduction in complex IV activities and levels of mitokines and regulatory RNAs are perturbed. Based on the cumulating evidence of mitochondrial abnormality as a disease contributor, it is therefore warranted to regard IBM as a mitochondrial disease, offering a feasible therapeutic target to be developed for this yet untreatable condition.

## 1. Introduction

Sporadic inclusion body myositis (IBM) is a rare condition, yet the most common inflammatory muscle disease in patients over 50 years old, with an annual incidence between 1 and 8 per million [1]. Most patients present initially with slowly progressive weakness of legs, arms, fingers and wrists. They report frequent falls or difficulty getting up out of a chair, weakened grip strength, some also experience difficulty swallowing. The disorder has a male predominance and is typically characterized by slowly progressive combined proximal and distal muscle weakness. Patient mobility decreases and patients develop fatigue due to declining muscle strength. Walking progressively becomes more difficult, which may render patients wheelchair-bound some 10 to 15 years after the onset of symptoms.

The muscle fibre degeneration displayed in IBM is more advanced than can be expected from the patient’s age. Muscle fibres contain the typical vacuoles rimmed with granular basophilic material and inclusions containing ubiquitinated modified proteins (Figure 1). Aggregates are composed of amyloid precursor protein-derived amyloid-β peptides, hyperphosphorylated tau, apolipoprotein E, α-synuclein and p62/Sequestosome 1 [2], clearly pointing to defective proteostasis as an underlying disease mechanism. In addition, endomysial inflammation accumulates and autoaggressive cytotoxic T-cells interact with human leukocyte antigen (HLA)-I-positive muscle fibres, creating foci of surrounded and invaded nonnecrotic myofibres [3]. In IBM patients, no significant clinical improvement can be gained by immunosuppressive treatments [4], which sparked the idea that the degenerative features of the disease are presumably more relevant clinically. However, the autoimmune nature of IBM has become more solid after the recognition of anti-cytosolic 5′-nucleotidase 1A (anti-cN1A) autoantibodies, which can be detected in 30% to 60% of patients [5]. It is becoming more and more clear that inflammation and degeneration are not separate entities in IBM and that inflammation may induce or aggravate the observed muscle degeneration. Sporadic inclusion body myositis is an acquired disease, yet genetic predisposition to IBM involving the Human Leukocyte Antigen—*DR* isotype β1 (HLA-DRB1) genotype has been described [6].

Understanding of mitochondrial health in these patients has been shadowed due to mitochondrial alterations receiving an amount of attention much inferior to degenerative and inflammatory changes. This review wants to turn that table, focusing in particular on mitochondrial alterations in IBM, revealing how mitochondrial abnormalities contribute to muscle degeneration and weakness in this as yet untreatable disease. The compelling histological, biochemical and molecular evidence of disrupted mitochondrial function is described hereunder.

## 2. Altered Mitochondrial Appearance

The first clues of mitochondrial dysfunction in IBM came from histological studies. Light microscopic evaluation of histochemical stains offers evidence of aberrant mitochondrial proliferation and functional defects [7]. Electron microscopy most conclusively reveals ultrastructural mitochondrial abnormalities, which include enlargement, loss of cristae of the inner mitochondrial membrane and paracrystalline inclusions. All these changes are, however, regarded as unspecific degenerative changes also associated with normal aging.

In addition, the distribution of mitochondria within the cells is altered in IBM muscle. Mitochondria are known to form a complex network that is constantly undergoing fusion and fission processes, allowing dynamic exchange of components. Mitofusins (MFN) mediate fusion of the outer mitochondrial membrane, while optic atrophy 1 (OPA1) governs fusion of the inner mitochondrial membrane [8]. Network disruption and increased mitochondrial fusion have been proposed in IBM. In evidence, MFN1 increases 1.7-fold (*p* < 0.05) at the protein and 4.2-fold (*p* < 0.001) at the messenger RNA (mRNA) level compared to control [9]. But another study reported OPA1 and MFN2 mRNA levels reduced 37% respectively. 31% [10]. Hence, complex disruption of the fusion/fission equilibrium, more so than upregulation of individual fusion components, could be at the heart of mitochondrial network disturbances.

## 3. Mitochondrial DNA Alterations

Mitochondrial DNA (mtDNA) sequence changes and clonal expansion of mtDNA deletions in individual muscle fibres [11,12] have been associated with IBM, though at relatively low abundance, and mtDNA depletion has repeatedly been described. Regardless of an increased mitochondrial content reported in muscle from 30 IBM patients, the amount of mtDNA dropped 36% in comparison to control tissues [10]. Another study confirmed that mtDNA copy numbers are significantly lower in IBM muscle than in controls [13]. These changes are thought to be due mostly to problems in mtDNA maintenance. Sporadic inclusion body myositis has also been shown to accumulate mtDNA deletions [13,14], of which the majority localizes to the direct repeats between nucleotides 8470 and 13447 [15]. In this respect, the mtDNA replication and maintenance factors come into view, including nuclear DNA-encoded DNA polymerase γ (POLG). DNA polymerase γ organizes the replication and repair of the mtDNA. Heterozygous POLG variants were present in 31% of IBM patients tested, with POLG variants more common in IBM patients with histological signs of mitochondrial deficiency [16]. Another important enzyme for mtDNA replication is the ribonucleoside-diphosphate reductase subunit M2B (RRM2B), which generates nucleotide precursors. Genetic defects in *RRM2B* lead to mtDNA depletion and IBM muscle displays higher variant frequencies than control muscle [16].

## 4. Mitochondrial Proliferation and Oxidative Phosphorylation Defects

Mitochondrial overload can be visualized with the Gomori trichrome stain, revealing the characteristic ragged red fibres under the microscope (Figure 2). This complex histological stain allows to differentiate connective tissue and fibrils (green) and nuclei (purple), from the endoplasmic reticulum (ER) and mitochondria (red). Due to massive subsarcolemmal accumulation of abnormal mitochondria, the contour of the muscle fibre appears irregular, causing a “ragged” aspect. Older compared to younger healthy subjects display 0.3 versus 0.02% ragged red fibres (*p* < 0.0001), while the frequency of ragged red fibres increases further to 1% in IBM patients. Frequencies in other inflammatory myopathy patients are similar to age-matched normal control subjects [17]. Ragged red fibres are often abundant in myoclonic epilepsy with ragged red fibres (MERRF), a mitochondrial disease caused by mutations in the mtDNA, of which the MT-TK gene is most commonly affected. Ragged red fibres are also prevalent in mitochondrial myopathy, encephalopathy, lactic acidosis and stroke-like episodes (MELAS) and Kearns–Sayre syndrome (KSS). In the ragged red fibres of heteroplasmic mtDNA defects, a high percentage of mutant genomes is found [18], suggesting that mitochondrial defects are the cause of the abnormal mitochondrial proliferation, reflecting a failed attempt to compensate activity loss. This compensatory increase of the cell’s mitochondrial load seems also to occur in IBM. In evidence, gene expression of the master regulator of mitochondrial biogenesis, peroxisome proliferator-activated receptor γ coactivator-1 α (PGC-1α), is significantly higher in IBM compared to control muscle [19].

Mitochondria are the main site of energy production in the cell. The process of oxidative phosphorylation (OXPHOS) uses electrons, gained by substrate oxidation through a chain of multiprotein complexes, to pump protons across the inner mitochondrial membrane. A proton gradient is generated and its force is used by the fifth and last OXPHOS complex to produce adenosine triphosphate (ATP). Perturbed OXPHOS activity can lead to ATP shortage, hence limiting energy consuming cellular processes. This results in excessive reactive oxygen species (ROS) generation and subsequent oxidative stress. Focal defects in OXPHOS are conspicuous in IBM muscle tissues. Scattered cytochrome c oxidase (COX) deficient fibres, although variable in amount, can be recognized in virtually all patient tissues [20]. These fibres with reduced COX (the fourth OXPHOS complex) activity, usually display normal succinate dehydrogenase (the second OXPHOS complex) staining. The percentage of COX deficient fibres correlates with the relative amount of mtDNA deletions [16]. Enzymatic studies confirm COX defects, detecting 30% decreased activities in IBM muscle when normalized to citrate synthase activity [10]. This phenomena appears an exaggerated form of the decay associated with normal muscle aging.

Sporadic inclusion body myositis is characterized by important changes in the proteome and transcription of genes associated with OXPHOS and mitochondrial function is affected in patients. This deregulation is illustrated by widespread changes in mRNA expression profiles as compared to healthy muscle [21]. In addition, the profile of untranslated RNAs is altered as well. Changes in levels of non-coding microRNAs (miRNAs) are recognized, the latter are important regulators of gene transcription, translation and mRNA turnover. In IBM muscle, miR-1 and miR-133 levels are markedly reduced [22]. miR-1 has been described to enter the mitochondria and stimulate translation of mtDNA-encoded transcripts [23]. miR-133-deficient mice display disturbed mitochondrial biogenesis, hence lower mitochondrial mass, leading to exercise intolerance [24]. Another class of regulatory RNAs are the long non-coding RNAs. Upregulation of long non-coding RNA 19 (H19), long non-coding myogenic differentiation antigen (lncMyoD), nuclear enriched abundant transcript 1 (NEAT1), plasmacytoma variant translocation 1 (PVT1), maternally expressed gene 3 (MEG3) and metastasis associated lung adenocarcinoma transcript 1 (MALAT1) have been reported in IBM muscle [21]. Several of these long non-coding RNAs are regulators of mitochondrial function. H19 and MALAT1 have been firmly linked to mitochondrial apoptotic pathways [25]. Mitochondrial stress alters NEAT1 expression resulting in nuclear retention of mitochondrial proteins and vice versa, NEAT1 depletion disrupts mitochondrial dynamics [26].

## 5. Dysfunctional Mitophagy

Cellular maintenance requires the elimination of dysfunctional elements and aberrant mitochondria do not escape surveillance by this removal-of-the-unwanted program. The disintegration of mitochondria by autophagy termed mitophagy, is initiated when unc-51-like autophagy activating kinase 1 (ULK1) comes into action, recruiting the Phosphatidylinositol-4,5-bisphosphate 3-kinase (PI3K III) nucleation and Phosphatidylinositol 3-phosphate (PI3P) binding complexes. The first contains a set of proteins that include Beclin1 and B-cell lymphoma 2 (Bcl2), the latter contains several autophagy-related proteins (ATGs). Autophagy-related (ATG) proteins, in a joint effort with 1A/B-light chain 3 (LC3), create autophagosomes that engulf targeted mitochondria and fuse with lysosomes. Sporadic inclusion body myositis is strongly associated with abnormal clearance and degradation of damaged mitochondria, hence the abundant changes to autophagy/mitophagy markers. Increased levels of the Bcl2 family protein B-cell lymphoma 2/adenovirus E1B 19kD-interacting protein 3 (BNIP3) can be shown both at the protein (2.5-fold, *p* < 0.05) and mRNA (3.2-fold, *p* < 0.01) level. Also, physical association of BNIP3 with LC3 [9] and accumulation in the aggregates [27] has been evidenced. Inefficient mitophagy in IBM muscle, leading to accelerating mitochondrial dysfunction, points to the tissue’s inability to perceive and respond adequately to the rampant mitochondrial stress underlying the disease.

## 6. Inflammation and Mitokines

Mitochondrial dysfunction and inflammation are linked via the activities of cytokines. The pro-inflammatory cytokines interleukin (IL)1-β and tumor necrosis factor α (TNF-α), which are in IBM abundantly present, are potent regulators of mitochondrial function [28]. In a cohort of 16 IBM patients, significant association of mitochondrial abnormalities with severity of inflammation and muscle fibre atrophy was noted [29], further suggesting a causative link between mitochondrial dysfunction and inflammation. Tumor necrosis factor -α-induced oxidative stress opens the membrane permeability transition pore, which leads to the uncontrolled transport of substances in and out of the mitochondria [30]. Pro-inflammatory cytokines further hamper tissue regeneration by suppressing the transcription of myogenic miRNAs, including miR-1 and miR-133a/b [22].

Mitokines are diffusible molecules released from cells in response to mitochondrial stress. A subgroup of mitokines are derived from the mtDNA and include humanin, an exercise-responsive peptide encoded by the *MT-RNR2* gene [31]. Another class of mitokines are metabolic cytokines, of which fibroblast growth factor 21 (FGF-21) is the best studied. Fibroblast growth factor 21 regulates energetic metabolism and expression levels are elevated in mitochondrial diseases [32]. Secretion of FGF-21 is induced during mitochondrial stress via the stress-activated transcription factor 4 (ATF4). Fibroblast growth factor 21 produced by muscle cells has been described to promote mitochondrial biogenesis and has positive regulatory effects on the glucose and lipid metabolism [33]. Normal [14,34] or only slightly increased [32] FGF-21 values have been reported in IBM sera. Another study reported increased FGF-21 in IBM plasma but the increase did not reach significance [10]. Growth differentiation factor 15 (GDF-15) is a member of the transforming growth factor β (TGFβ) family and another regulator of energy homeostasis and potent promoter of oxidative metabolism and lipolysis [35]. Growth differentiation factor 15 levels are increased in patients with mitochondrial defects, hence it has been put forward as a valuable biomarker for oxidative phosphorylation deficiencies [36]. Our own studies recently revealed an increase of GDF-15 levels in IBM sera (Figure 3), suggesting this mitokine as a novel biomarker for this disorder as well.

## 7. Mitochondrial Defects in other Muscle Diseases

Mitochondrial dysfunction is not unique to IBM but has been shown to associate also to varying degrees with other idiopathic inflammatory myopathies [37]. A subset of patients with polymyositis have been described to display fulminant mitochondrial abnormalities, forming an IBM-like syndrome designated as polymyositis with mitochondrial pathology (PM-Mito). However, the diagnosis of polymyositis in these patients is contested by many experts, as they regard subsequent response to immunosuppressive therapy as a diagnostic criterion. Similar to IBM, PM-Mito patients are most often poorly responsive to corticosteroid treatment [38]. A cohort of patients with anti-3-hydroxy-3-methyl-glutaryl-coenzyme A reductase (HMGCR) autoantibody positive immune-mediated necrotizing myopathy also show clear signs of mitochondrial degeneration, with loss of cristae in the vast majority and mitophagy with abundant autophagic vacuoles often containing remnants of mitochondria [39]. The latter study found mitochondrial aberrations at a higher frequency even than in the included IBM controls. In evidence, miR-1 and miR-133 reduction was also found in other inflammatory myopathies, although the decrease remained more profound in IBM. Transcriptome analysis in dermatomyositis revealed a major cluster of downregulated genes related to mitochondrial activities, translating into mitochondrial abnormalities, increased ROS and decreased OXPHOS [40]. In addition, the prominently affected perifascicular areas of dermatomyositis muscle are also the site where most mitochondrial alterations can be observed [41]. Mitochondrial damage has also been observed in medication-related muscle disorders. Statin-induced myopathy is associated with mitochondrial dysfunction [42] and mtDNA depletion [43]. The effect statins exert on mitochondrial OXPHOS capacities could be established by a randomized controlled trial, that showed a reduction of respiratory chain enzyme and citrate synthase activities [44]. It appears that the crucial role played by mitochondria in the energy-demanding skeletal muscle tissue makes them highly vulnerable to dysfunction.

## 8. Consequence for Therapeutic Intervention

Patients with IBM are poorly responsive to classical immune-suppressants including prednisone. Also, clinical improvement under other immunomodulatory drugs such as intravenous immunoglobulin G (IgG), methotrexate, the T-cell depleting monoclonal antibody alemtuzumab are generally disappointing [3]. Alternative routes for therapy are therefore badly needed.

An anti-myostatin approach has been tested in the RESILIENT trial but did not improve the patients’ performance in the 6 min walk test and did not increase muscle strength [45]. The benefits of physical activity to IBM have, however, generated more hopeful results. Exercise helps to maintain muscle function in general, improves aerobic capacity [46] and increases muscle strength. Exercise training stimulates not only the biogenesis of mitochondria but also stimulates the removal of old and unhealthy mitochondria through mitochondrial dynamics and autophagy [47]. The positive effects of exercise are multifaceted but mitochondrial biogenesis may be controlled primarily by targeted activation of peroxisome proliferator-activated receptor γ coactivator 1-α (PGC-1α) via p38 mitogen-activated protein kinase activation [48].

Therapies supporting mitochondrial activities represent an attractive novel route in IBM and compounds that protect mitochondrial integrity are an alternative therapeutic avenue to be explored. Anti-oxidants and vitamins, co-factors and nutritional supplements can be administered, for which vitamin C, vitamin K, thiamine, folic acid, l-carnitine and creatine are plausible candidates. Though these approaches usually lack a major therapeutic impact in primary mitochondrial disease [49], they might still be of benefit to IBM patients. Another approach could focus on restoring or bypassing defective mitochondrial components. The use of supplements to treat mitochondrial disease caused by defective biosynthesis of coenzyme Q10 and riboflavin has been tried. In addition, compounds that increase mitogenesis are a possibility. Pioglitazone, which was approved by the U.S. Food and Drug Administration (FDA) in 1999 for treating diabetes, was shown to improve mitochondrial function in the skeletal muscles of diabetic patients. An open-label pilot study of pioglitazone in 15 IBM patients is currently undertaken at Johns Hopkins University (Baltimore, MD, USA), to see if PGC-1α levels and muscle strength can be increased by the treatment regimen.

## 9. Conclusions

Muscle of IBM patients seems to display exaggeration of normal aging-associated degenerative changes, which includes mitochondrial decline. The clinical importance of mitochondrial dysfunction is difficult to evaluate, as it cannot be separated from the other pathological changes that occur in IBM muscle. However, it appears the disease-related fatigue and exertion in IBM patients reflects their mitochondrial impairment [50]. In many aspects, the mitochondrial dysfunction of IBM resembles findings in primary mitochondrial myopathies and changes found in the blood metabolome are strikingly similar [14]. We can thus conclude that, although mitochondrial alterations are not the genetic origin, they nonetheless represent an important aspect of IBM disease mechanisms and represents a druggable and valid therapeutic target.

## Figures and Tables

**Figure 1 biomolecules-09-00015-f001:**
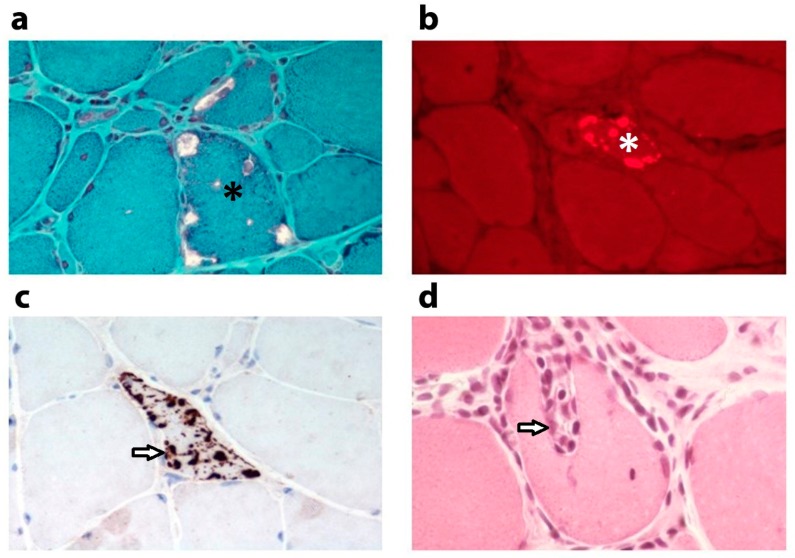
Histological features of sporadic inclusion body myositis muscle. Histological characteristics in a quadriceps muscle biopsy from a male 73 years old patient diagnosed with sporadic inclusion body myositis. (**a**) Modified Gomori trichrome stain reveals several vacuoles in a muscle fibre (asterisk). (**b**) Congo red stain detected under fluorescent light visualizes a muscle fibre with inclusions containing β-amyloid (asterisk). (**c**) Immunostaining of the ubiquitin-binding scaffold protein and autophagy receptor p62/sequestosome1 (3’-Diaminobenzidine stain, brown) shows sarcoplasmic p62-immunoreactive aggregates in a muscle fibre (arrow). (**d**) Haematoxylin and eosin stain showing autoaggressive inflammatory cells targeting a nonnecrotic muscle fibre (arrow). Magnification ×785 before reduction.

**Figure 2 biomolecules-09-00015-f002:**
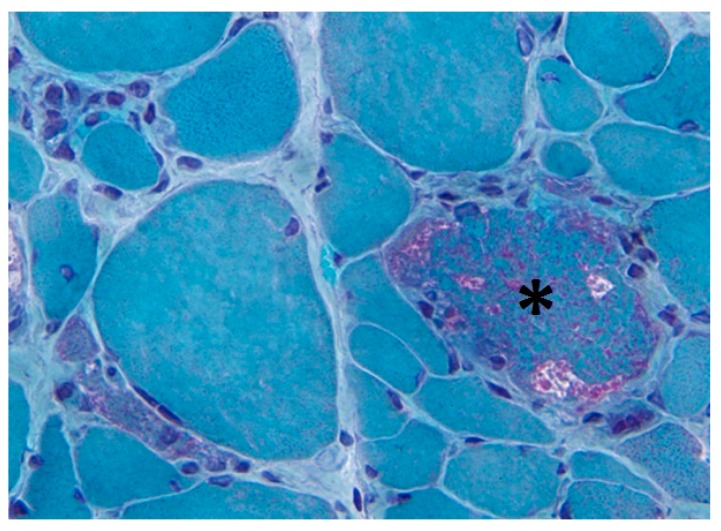
Modified Gomori trichrome stain showing a ragged red fibre. A ragged red fibre (asterisk) is present in the quadriceps muscle biopsy of a 67 years old female sporadic inclusion body myositis patient. Magnification ×785 before reduction.

**Figure 3 biomolecules-09-00015-f003:**
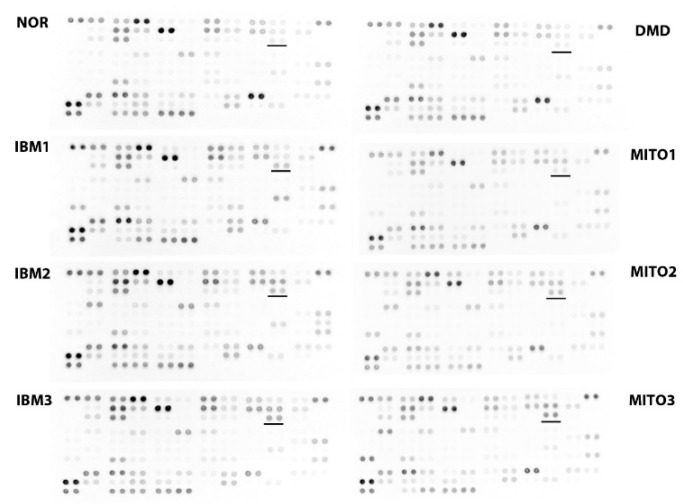
Serum growth differentiation factor 15 (GDF-15) levels in patients with sporadic inclusion body myositis. The Human XL Cytokine Proteome Profiler Array (Bio-Techne, Abingdon, UK) visualizes double spots representing GDF-15 levels in serum (underlined). NOR: Normal commercial control sample (Sigma, Overijse, Belgium); IBM1–3: Patients diagnosed with sporadic inclusion body myositis—IBM1 (female of 70 years); IBM2 (male of 72 years); and IBM3 (male of 67 years); DMD: Patient with Duchenne muscular dystrophy due to *DYS* deletion of exons 48–50 (male of 9 years); MITO1–3: Patients diagnosed with primary mitochondrial diseases—MITO1 (patient with homoplasmic *MT-ND4* mutation; male of 49 years); MITO2 (patient with heteroplasmic *MT-TL1* mutation; female of 54 years); and MITO3 (patient with *POLG* mutation; female of 51 years).

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
