# Peer review of "Sporadic Inclusion Body Myositis: An Acquired Mitochondrial Disease with Extras"

_biomolecules, 2019, doi:10.3390/biom9010015_

Round 1
Reviewer 1 Report
A timely and well written manuscript. I make the following suggestions to improve clarity:
· Line 25, page 1: The reference provided does not provide any primary evidence to support the assertion that IBM is the most common IIM in the over 50s, as far as I can tell. Please update with a reference citing primary research supporting this assertion.
· The caption to Figure 1 (Line 54) refers to “Congored”. I think this should read “Congo red”.
· Line 35, page 1, reference 3: I would prefer if references to original research, or at least a review specifically about previous failed clinical trials in IBM, were provided to support the assertion that immunosuppressive treatments do not work.
· Line 38, page 1, reference 4: Similarly, it would be preferable to cite the original papers describing CN1A, rather than secondary sources.
· Line 45, page 2: I don’t really agree that mitochondrial change is seen as “less significant” in IBM pathogenesis. Would suggest removal of this sentence, or update to provide a citation supporting the assertion.
· Section 3 (page 3, line 76): I suggest inclusion and discussion of the following important reference: Rygiel, et al, https://www.ncbi.nlm.nih.gov/pubmed/27131788
· Section 4 (first sentence, line 93): this would benefit greatly from inclusion of a picture demonstrating a ragged red fibre. Can this be included please?
· Line 193: The description of the PM-Mito phenotype is important, but appears to contradict a previous assertion on line 99 (section 4) that frequencies of ragged red fibres in non-IBM IIMs is similar to age-matched normal controls. I would suggest adding further explanation here to improve clarity.
· Line 207, page 6: The authors say “could be established by a randomised controlled trial”, but then reference an RCT which has already been performed. Suggest rewording to highlight the outcomes from this already completed trial.
· Table 1 seems unnecessary in its current form. I suggest updating to make more relevant to the topic of the review – i.e. a summary of mitochondrial change/pathology in various muscle diseases including IBM
· Line 216 – see comment above regarding reference 3
Author Response
Answers to the remarks of Reviewer 1
A timely and well written manuscript. I make the following suggestions to improve clarity:
· Line 25, page 1: The reference provided does not provide any primary evidence to support the assertion that IBM is the most common IIM in the over 50s, as far as I can tell. Please update with a reference citing primary research supporting this assertion.
The reference has been changed to Dimachkie, M.M.; Barohn, R.J. Inclusion body myositis. Semin Neurol 2012, 32, 237-245, which contains epidemiological details.
· The caption to Figure 1 (Line 54) refers to “Congored”. I think this should read “Congo red”.
The mistake has been corrected.
· Line 35, page 1, reference 3: I would prefer if references to original research, or at least a review specifically about previous failed clinical trials in IBM, were provided to support the assertion that immunosuppressive treatments do not work.
The reference has been changed to Breithaupt, M.; Schmidt, J. Update on treatment of inclusion body myositis. Curr Rheumatol Rep 2013, 15, e329, a paper that summarizes immune-modulatory treatment outcome in patients.
· Line 38, page 1, reference 4: Similarly, it would be preferable to cite the original papers describing CN1A, rather than secondary sources.
The original publication Pluk, H.; van Hoeve, B.J.; van Dooren, S.H.; Stammen-Vogelzangs, J.; van der Heijden, A.; Schelhaas, H.J.; Verbeek, M.M.; Badrising, U.A.; Arnardottir, S.; Gheorghe, K.; Lundberg, I.E.; Boelens, W.C.; van Engelen, B.G.; Pruijn, G.J. Autoantibodies to cytosolic 5'-nucleotidase 1A in inclusion body myositis. Ann Neurol 2013, 73, 397-40, is now cited as reference 4.
· Line 45, page 2: I don’t really agree that mitochondrial change is seen as “less significant” in IBM pathogenesis. Would suggest removal of this sentence, or update to provide a citation supporting the assertion.
I agree with the reviewer that this was too strongly put. The text now reads: Understanding of mitochondrial health in these patients has been shadowed due to mitochondrial alterations receiving an amount of attention much inferior to degenerative and inflammatory changes
· Section 3 (page 3, line 76): I suggest inclusion and discussion of the following important reference: Rygiel, et al, https://www.ncbi.nlm.nih.gov/pubmed/27131788
Indeed, I agree with the reviewer and changed reference 11 to Rygiel KA, Tuppen HA, Grady JP, Vincent A, Blakely EL, Reeve AK, Taylor RW, Picard M, Miller J, Turnbull DM. Complex mitochondrial DNA rearrangements in individual cells from patients with sporadic inclusion body myositis. Nucleic Acids Res 2016, 44, 5313-29. Also, the text now mentions clonal expansion of mtDNA deletions.
· Section 4 (first sentence, line 93): this would benefit greatly from inclusion of a picture demonstrating a ragged red fibre. Can this be included please?
A new figure 2, representing a ragged red fiber, is now included.
· Line 193: The description of the PM-Mito phenotype is important, but appears to contradict a previous assertion on line 99 (section 4) that frequencies of ragged red fibres in non-IBM IIMs is similar to age-matched normal controls. I would suggest adding further explanation here to improve clarity.
Diagnosing this subgroup of PM-Mito patients is controversial. PM is by many experts defined as myositis with endomysial inflammation and invaded nonnecrotic fibers, without signs of degeneration and aggregation, and that had reacted to immunosuppressive treatment. The text has been altered to include this discussion.
· Line 207, page 6: The authors say “could be established by a randomised controlled trial”, but then reference an RCT which has already been performed. Suggest rewording to highlight the outcomes from this already completed trial.
The results of this trial is now included in the text.
· Table 1 seems unnecessary in its current form. I suggest updating to make more relevant to the topic of the review – i.e. a summary of mitochondrial change/pathology in various muscle diseases including IBM
As both reviewers find this table redundant, the table has been omitted.
· Line 216 – see comment above regarding reference 3
The reference has been replaced.
Reviewer 2 Report
Comments:
Introduction:
line 33: The author states that there is accumulation of endomysial inflammation. Progressive inflammation has not been shown in IBM. What does he mean/imply?
line 33: Autoaggressive macrophages invade non-necrotic muscle fibers. I believe this has not been shown for IBM. If so, please provide a reference and explain why this would be the case.
line 42/43: Mentioning hereditary IBM is confusing as it is a different disorder, and selective, as other IBM's are not mentioned such as IBM-FTD-Paget. Please remove.
section 2
line 60/61: What are the light microscopic changes that suggest mitochondrial dysfunction?
Table 1: redundant, as this paper is on IBM. A reference to, for example ENMC, criteria should suffice.
section 7
line 242: disease related fatique and exertion pops up in the conclusion, but should be addressed earlier on in the paper. What is the evidence for the presence of fatique in IBM?
Author Response
Answers to the remarks of Reviewer 2
Introduction:
line 33: The author states that there is accumulation of endomysial inflammation. Progressive inflammation has not been shown in IBM. What does he mean/imply?
I apologize for this confusing sentence, which was caused by wrongful reference to Figure 1 at this point. The figure does not show inflammation, but highlights the vacuoles and aggregates in muscle fibers. This has been corrected.
line 33: Autoaggressive macrophages invade non-necrotic muscle fibers. I believe this has not been shown for IBM. If so, please provide a reference and explain why this would be the case.
New reference 3 has been added, and the text has been changed to better reflect the aspects of inflammatory reactions in IBM.
line 42/43: Mentioning hereditary IBM is confusing as it is a different disorder, and selective, as other IBM's are not mentioned such as IBM-FTD-Paget. Please remove.
This entry and the corresponding reference have been removed.
section 2
line 60/61: What are the light microscopic changes that suggest mitochondrial dysfunction?
These changes are discussed further on, under section 3, and the manuscript now explains this.
Table 1: redundant, as this paper is on IBM. A reference to, for example ENMC, criteria should suffice.
As both reviewers find this table redundant, the table has been omitted.
section 7
line 242: disease related fatique and exertion pops up in the conclusion, but should be addressed earlier on in the paper. What is the evidence for the presence of fatique in IBM?
Fatigue is now explained in the introduction as an important clinical sign of IBM.